# Efficient Cryopreservation of *Populus tremula* by In Vitro-Grown Axillary Buds and Genetic Stability of Recovered Plants

**DOI:** 10.3390/plants10010077

**Published:** 2021-01-02

**Authors:** Elena O. Vidyagina, Nikolay N. Kharchenko, Konstantin A. Shestibratov

**Affiliations:** 1Branch of the Shemyakin-Ovchinnikov Institute of Bioorganic Chemistry of the Russian Academy of Sciences, Science avenue 6, Pushchino, Moscow Region 142290, Russia; vidjagina@mail.ru; 2Voronezh State University of Forestry and Technologies Named after G.F.Morozov, 8 Timiryazeva Str., Voronezh 394087, Russia; forest.vrn@gmail.com

**Keywords:** aspen, transgenic lines, cryopreservation, vitrification

## Abstract

Axillary buds of in vitro microshoots were successfully frozen at –196 °C by the one-step freezing method using the protective vitrification solution 2 (PVS2). Microshoots were taken from 11 transgenic lines and three wild type lines. Influence of different explant pretreatments were analyzed from the point of their influence towards recovery after cryopreservation. It was found out that the use of axillary buds as explants after removal of the apical one increases recovery on average by 8%. The cultivation on growth medium of higher density insignificantly raises the regenerants survival rate. Pretreatment of the osmotic fluid (OF) shows the greatest influence on the survival rate. It leads to the increase in survival rate by 20%. The cryopreservation technology providing regenerants average survival rate of 83% was developed. It was based on the experimental results obtained with explant pretreatment. Incubation time in liquid nitrogen did not affect the explants survival rate after thawing. After six months cryostorage of samples their genetic variability was analyzed. Six variable simple sequence repeat (SSR) loci were used to analyze genotype variability after the freezing-thawing procedure. The microsatellite analysis showed the genetic status identity of plants after cryopreservation and of the original genotypes. The presence of the recombinant gene in the transgenic lines after cryostorage were confirmed so as the interclonal variation in the growth rate under greenhouse conditions. The developed technique is recommended for long-term storage of various breeding and genetically modified lines of aspen plants, as it provides a high percentage of explants survival with no changes in genotype.

## 1. Introduction

The extensive application of biotechnology, genetic engineering, genome editing and genome breeding in modern agriculture and forestry leads to creation of a wide variety of unique genotypes (breeding and hybrid forms, transgenic lines, candidate varieties and varieties) [1,2,3]. The preservation of these unique genotypes in their original state is very important throughout the entire period of breeding work, from the first hybridization or genetic transformation to subsequent commercialization [4]. One of the effective solutions for this task is to create in vitro collections [5]. We also need to take into consideration the fact that conservation of unique genotypes via regular in vitro subcultivation requires not only significant expenses, but also has the risk of infecting the culture, accumulating somaclonal variations and, ultimately, may lead to the loss of genotype [6,7]. This is the main reason why different ways of preserving unique genotypes are currently being considered. The most promising method is cryopreservation of plant material in liquid nitrogen at a temperature of –196 °C, when all the processes in cell stop and plant material can be stored unchanged for a theoretically unlimited time [8,9,10,11,12]. This method was also successfully applied with woody plants [4,13,14,15].

In vitro culture is most acceptable for conservation of modified lines [1,16]. It is so because transformants are obtained in vitro conditions and initial screening is performed in the same conditions [17]. However, cryopreservation of such cultures has a number of problems, as the plant material contains a lot of moisture and is sensitive to drying out. In addition, microshoot tissue type selection is also an important and a very ambiguous moment for different types of cultures [18,19].

The PVS2-based vitrification developed by Sakai et al. [20] is the most acceptable method for cryopreservation of in vitro cultures of woody plants. This method is the «one-step freezing method», which allows to place cryotubes with explants directly into the liquid nitrogen (LN) without gradually decreasing the temperature [5]. It helps to reduce cost significantly and to improve the survival rate of explants due to a decrease in the effect of cryoprotectant usage and lower formation of ice structures [21]. Using the shoot tips is mostly preferable for woody plants [5,14,22], as helps to preserve the genotype of plants successfully and speeds up the process of culture recovery. Despite of many variants in the plant cryopreservation area, there is no common cryopreservation protocol suitable for all the woody plant cultures [5]. It is connected with the peculiarities of culture: moist and size of tissue cells, physiological peculiarities, recovery rate and multiplication [5,21,23].

Plants of the genus *Populus* are the main model woody plants for various genetic manipulations. Though, this model plant is rather difficult to preserve ex vitro and in vitro collections due to a number of biological features. The article by Tsai and Hubscher [24] draws attention to the fact that the development of an economically valid strategy for the long-term conservation of wild type and transformed tree lines is very important part of *Populus* functional genomics future research. Nowadays there is a lot of research works in creating transgenes based on the *Populus* genus plants. [25]. Earlier several dozen lines were analyzed for one construction. Now with the development of genomic selection [26], it becomes possible to screen a large amount of data, and it allows to process hundreds and thousands of lines to search for the most promising combinations. Regarding to this, problem of saving the maximum number of breeding lines is quite crucial. It is also important to save not only modified plants but the ones used in research: pedigreed lines [27] and wild hybrids with known ecophysiological characteristics [28].

That is why, work on creating cryopreservation protocols of the *Populus* genus plants lasts for many years, and significant success is achieved [4,14,15,23], but not all problems were solved so far. For in vitro plant material, a much larger than for plant material ex vitro variation in explant survival was observed [4]. The probable reasons of such difference can be biochemical and morphological features of cells in vitro culture. It is noted that the in vitro plant survival rate depended on the methods of explant pretreatment before immersion in liquid for cryopreservation [24]. It is proposed to make preliminary cultivation at low temperatures for plant explants pretreatment [14,15,23] and to add sugars as osmotic agents [14,23]. However, it was found out that conditions for pretreatment and storage should be selected individually for each specific genus and genotype [5]. It is also worth mentioning that the problem of genetic stability saving in plants which undergo cryopreservation is not completely resolved [19] and there are not so many works published on this issue at the moment. There are several works that consider the issue of preserving the plant genotype after long-term cryopreservation [4,29,30,31], but the results obtained do not give a complete understanding of the possible effect of different cryopreservation methods on the genetic component when used in other plant species.

Therefore, elaboration of new approaches for the *Populus* genus plants (transformed and wild valuable genotypes) long-term storage without changing their genetic status is an urgent and timely task which can help to reveal new opportunities in woody plants genomic analysis. Regarding to this, we considered in our article possible ways to increase the regenerate survival rate using the example of aspen transformed plants and wild types after the freezing—thawing procedure and storage in liquid nitrogen by changing the explant preparation conditions, and taking into account preservation of recovered plants genetic stability.

## 2. Results

### 2.1. Analysis of the Pretreatment Effect on the Regenerant Survival Rate after Cryostorage

To determine possible influence of various factors on the explant survival after the freezing - thawing procedure, experiments with different pretreatments were laid down. In vitro cryopreservation of microshoot axillary buds by rapid freezing in the presence of PVS2 was successful with all pretreatment options. After the thawing process, explant differentiated tissues of all experimental options died off, and only the axillary bud remained viable ensuring restoration (Figure 1A). Micrographs in Figure 1A present the Pt genotype explants one week after thawing and transfer to the recovery medium; explants of other genotypes were identical (data not presented). Further analysis of the three-week-old regenerants showed that no callus formation was observed with all experimental plants obtained, and recovery proceeded directly from the axillary bud (Figure 1B).

Regenerants survival varied depending on the pre-treatment selected. The most reliable differences affecting survival were registered in the experiment using the OF pretreatment (Figure 2a). The average survival without the OF pretreatment was 57%, survival of plants with the osmotic fluid (OF) treatment was 72%. It was also noted that using axillary buds after removal of the apical bud led to an increase in the regenerant survival by 8%, but the survival did not exceed 61% (Figure 2b). Introduction of denser medium with content 1% agar-agar insignificantly (by an average of 4%) increased regeneration; however, increasing medium density up to content 1.2% agar-agar either did not have any significant effect, or adversely affected survival rate of Pt and f2 genotype explants (Figure 2c).

Further analysis of microshoots obtained from regenerants after 8 weeks of incubation on the recovery medium showed that visually distinguishable morphological alterations after freezing-thawing were not found in all types of experiments (Figure 1C). Figure 1C demonstrates typical representatives of the Pt genotype regenerants; for the rest of the genotypes, recovery was similar (data not presented). Shoots successfully laid lateral buds also without formation of callus. Preparing an experiment on plant rooting to assess alterations in the genetic component under greenhouse conditions showed that the obtained regenerants did not have deviations in root formation under the in vitro conditions.

### 2.2. Analysis of the Aspen Transformed Genotypes Survival after Long-Term Cryopreservation

Most favorable conditions were selected for transgenic aspen lines cryopreservation, which showed an increase in the regenerate survival on non-transformed Pt, f2 and PtV22 genotypes. Non-transformed plants were also included in the experiment. Microplants cultivated on growth medium with 1% agar-agar were used, four-day-old axillary buds were extracted after removal of apical buds, and then the explants were treated with OF. Further, explants were also transferred to PVS2 and stored in LN. After 3 and 6 months of cryostorage, explants were thawed and transferred to recovery medium. Complex action approach in the explant pretreatment made it possible to significantly increase the regenerant survival after the freezing-thawing procedure. Survival rate was 83% on average, which was significantly higher than all previously obtained survival rates in all types of experiments (Figure 3). Statistically significant differences in survival between wild types and transformed plants were not observed; however, survival rate for lines carrying the recombinant *sp-Xeg* gene was lower than for other transgenic and wild plant genotypes by an average of 7%. No significant relationship was found between the storage time in LN and the regenerants survival rate after the freezing - thawing procedure.

### 2.3. Assessment of Clonal Fidelity after Cryopreservation

To determine integrity of genotypes restored after the plant cryostorage, analysis was initially carried out on the presence of recombinant insertion of the plant transgenic genotypes: PtXIBar9a, PtXIBar14a, f2XIBar5a, PtV22XBar2a, Xeg-1-1a, Xeg-1-1c, Xeg-2-1b and Xeg-2-3b. PCR analysis showed presence of the recombinant gene introduction with required size in all transgenic lines (Figure 4).

Genetic marking was performed using SSR markers at 6 loci for all experimental lines. Fragmentary analysis demonstrated that genetic profiles of plants that were exposed to the freezing-thawing procedure and of plants that were not exposed to these manipulations were identical (Table 1).

Analysis of plants under greenhouse conditions showed that the data on height of cryopreserved plants and plants without cryopreservation were similar (Figure 5). Morphological deviations were not found in lines that were exposed to the freezing-thawing procedure. Transformed plants showed interclonal variation from 46 cm to 64 cm. Xeg-1-1a and Xeg-2-1b lines, which exceeded the control by about 30%, as previously noted, appeared to be fast-growing [32]. Thus, transformed lines retained their phenotypic characteristics after the cryopreservation process.

## 3. Discussion

Due to the fact that *Populus* plants in the forest biotechnology are the main model plants, all variations of the obtained and wild genotypes preservation became an important problem. It was proven that aspen subcultivation in vitro leads to alterations in genetic material without visible external signs of somaclonal variability [33]. Based on this, usage of cryopreservation technologies with plants of this genus gains major importance. The main criteria in selecting one or another cryopreservation technology are preservation of the genetic component and high percentage of the explant survival after storage. However, the problem of selecting conditions for cryopreservation of *Populus* genus plants remains unresolved. Studies show that this genus regenerated plants obtained from in vitro culture survival rate varies greatly from 3% to 93% [24]. This is why it is necessary to unify the developed technologies via identifying key points for the survival rate improvement. Moreover, the works of recent years show that it is important not only to increase the survival rate of regenerants, but also to pay attention to the preservation of their original genotype [4,29,30,31]. This can make the cryopreservation process more efficient [29]. In this regard, we have suggested and experimentally verified the idea that pretreatment of explants for cryostorage can subsequently significantly increase the survival rate. This approach can guarantee the safety of genetic information in aspen plants.

Studies conducted on tops of the white poplar shoots [14] showed that prolonged cryopreservation solution exposure with gradual cooling could damage the experimental material. This is why, we applied a new approach to cryopreservation of plants in our study. This is the one-step freezing method based on cryoprotective vitrification. This method of cryopreservation allows to transfer tubes with explants directly to the LN, it helps to inhibit ice formation and reduces the duration of the toxic effect of cryostorage solutions. To optimize the one-step freezing method, we tested various options in pretreatment of explants for PVS2 processing and further storage. It was proved that pretreatment of explants is crucial in successful implementation of this method [4]. Using the most successful combinations made it possible to develop technology in aspen axillary bud cryopreservation with the average survival rate of 83%, which is a very high indicator [34]. Our studies demonstrated that the difference in regeneration for certain transgenic lines, such as transformants carrying the *sp-Xeg* gene, could be significantly lower in comparison with the others. This may be due to the peculiarity of the recombinant gene action, which leads to an increase in the cell size [35], as a result, it is able to significantly affect the cryoprotector action efficiency. Thus, it is rather important to take into account features of the recombinant insert in the transformed lines.

For the selected transgenic and non-transgenic lines, it was decided not to increase the number of passages, to reduce somaclonal variability. This is why cultivation on medium with a high sugar content to reduce the moisture level of the culture [14,36] was not suitable. Medium with the high agar-agar content is introduced in three passages before cryopreservation during the planned replanting and does not require any increase in the number of subcultivations. To stimulate cell vitrification and subsequent increase in the survival rate, cold hardening technique was used in the original microplants for 3 days at low positive temperatures [37,38,39]. Shoots 21 days of age were taken, as the most suitable for such manipulations [40].

Meristem structure is another important parameter in increasing the survival rate, as it is shown by Thinh et al. [41]. Fully covered meristems showed very low survival rate; however, completely exposed meristem tissue is vulnerable to detrimental effects of cryoprotectants which could lead to callus formation [41,42]. Thus, not the apical buds but smaller auxiliary buds were selected for cryopreservation of transgenic genotypes as the explants. Axillary buds are smaller in size and also less closed by leaf primordia; besides, their number on the shoot makes it possible to obtain more explants. It significantly increases probability of the culture successful recovery. Axillary bud’s usage proved this strategy to be correct. Microshoot was restored directly from the axillary bud without any callus formation. It helps to exclude possible variability. Tissues near an auxiliary bud died completely in the process of recovery. This is probably due to large size of the surrounding differentiated cells. Similar effect was already described earlier, when only the meristem cells survived, probably because of their small size [19,43]. To increase regeneration, it is important for bud and meristem size in them to be small enough. It allows them to withstand detrimental effect of the dying differentiated tissue cells and osmoprotectors. In this regard, we have proposed a suggestion: that swollen axillary buds would be more acceptable for cryopreservation. Our studies showed that the use of axillary buds after removing the apical bud could significantly increase (by 8%) the survival rate of a regenerant. To avoid toxic effect of the surrounding dying tissues, the size of the secreted enzyme including the axillary bud was not exceeding 2–3 mm. Similar works show that the explant size is directly connected to survival rate after cryopreservation [19].

The osmotic fluid (OF) pretreatment was used for additional dehydration of plant material. It showed an increase in the survival rate of regenerates by 20%. Osmotic fluid influences this way because of its hypertonic features. It allows to change the cell water content and prepares the cell to PVS2 solution action. The main feature of this mechanism is the following: it maximizes concentration of the cell solute molecules. It prevents formation of crystals from the remaining water molecules [44]. The freezing temperature becomes much lower to the eutectic point, which leads to solidification of the entire system [21]. This approach made it possible to increase the regenerate survival rate. It also helped to reduce the time of plant pretreatment without additional subcultivation. Moreover, an increase in the survival rate was achieved by using the mixture of penetrating and non-penetrating cryoprotectants. This helps to increase the survival rate after the freezing - thawing procedure significantly [21,45]. To increase percentage of regeneration, we used OF and PVS2 solutions. They contain DMSO and glycerol that penetrate the cell and additive that penetrates only through the cell wall—maltose; and non-penetrating additives—sucrose and ethylene glycol.

To recovery the culture we used rapid heating to thaw the explants; this improves the survival rate, as very small ice crystals could turn into larger and destructive ones, if the frozen system is being heated too slowly [46]. In addition, this approach was chosen because rapid heating at 45 °C could provide a quick transition from glass stage to liquid without passing the ice phase [21].

Using pre-cultivation on medium with higher gelling agent content; removal of apical buds and introduction of osmotic fluid in pretreatment of explants allowed to increase survival rate from 55% to 83%. So, we can state that only the combined effect of different factors is able to significantly increase the survival rate of regenerants. This is why this technology is recommended for in vitro cryopreservation of aspen plants. Our research also shows that storage duration in the developed technology does not affect the degree of regeneration recovery. We calculated that according to recommendations on creating a cryoreservoir we should take at least 50 explants per genotype for successful and safe storage [47].

We analyzed the safety of genetic characteristics after cryopreservation. The results showed that there were no alterations in the genotype. The recombinant genetic construct in transgenic plants so as genetic status during genotyping using the SSR markers remain undamaged. The selected marker loci have a high degree of variability. They are used to determine genetic changes within one specie and are also used to identify somaclonal variability [33,48,49]. They show the most insignificant changes in the genotype. The markers selected for genotyping are highly polymorphic, which indicates differences in the genetic profile of the lines studied by us before the cryopreservation process. For example, even transgenic lines obtained on the same wild genotype have different genetic profile and are easily identified. We showed that the cryopreservation process was not leading to any modification in the genotype. However, we did not conduct a genetic analysis of the non-regenerated explants. Probably, due to a possible change in the genotype, these explants did not survive [29]. Moreover, our data are confirmed by the data obtained in previously published works. When using PVS2-treated for in vitro buds of hybrid poplar, for which no genetic changes identified by the RAPD method were also found [4]. Jokipiiet et al. indicate that in addition to molecular methods, it is necessary to carry out morphological analysis to confirm the preservation of the genetic status. In this regard, we analyzed the plants that underwent the process of cryopreservation in a greenhouse. Analysis of growth indicators showed that there were no changes in morphological characteristics and height. At the same time, the transformed plants with the *sp-Xeg* gene showed saving of changes in the interclonal variation, which was noted earlier [32]. This confirms that the transgenic line characteristics after cryopreservation remain unchanged.

## 4. Materials and Methods

### 4.1. Plant Material

In the experiment, 11 genotypes of wild types of aspen and transformed lines obtained on their basis were used. Wild types of aspen: Pt and f2 (*Populus tremula L.*), and PtV22 (*Populus tremula x Populus tremuloides*). Plants with the Pt genotype are characterized by rapid growth and resistance to heartwood rot [35]; plants with the f2 genotype are distinguished by high density wood [50]. Plants with the PtV22 genotype are characterized by rapid growth and resistance to heartwood rot. Transgenic plant lines carried recombinant genes resistant to the phosphinothricin *bar* [50]: PtXIBar9a, PtXIBar14a, f2XIBar5a, PtV22XBar2a; and fungal xyloglucanase *sp-Xeg* [35]: Xeg-1-1a, Xeg-1-1c, Xeg-2-1b, Xeg-2-3b. Plants with non-transformed wild genotypes (Pt, f2, PtV22) were involved in experiments A, B and C. Based on the test results, an experiment was set up involving all non-transformed and transformed lines (Pt, f2, PtV22, PtXIBar9a, PtXIBar14a, f2XIBar5a, PtV22XBar2a, Xeg-1-1a, Xeg-1-1c, Xeg-2-1b and Xeg-2- 3b). Plants were cultivated in vitro on the Woody Plant Medium (WPM) [51] with the 3% sucrose, agar-agar 0.8% at 22–24 °C and photoperiod of 16 h with a light intensity of 2500-3000 lux.

### 4.2. Preculture Treatments, Vitrification Procedure and Rapid Cooling

In vitro cultures of non-transformed wild types aspen (genotypes Pt and f2, and PtV22) were divided into three groups for different experiments of explants’ pretreatment (A, B and C), which included different types of explants, cultivation on medium with different densities and osmotic treatments. Axillary buds of microshoots and axillary buds obtained after preliminary removal of the apical bud and cultivation for 4 days on the WPM growth medium, agar-agar 0.8%, were used as explants in experiment A. In experiment B, explants of axillary buds of microshoots were used obtained after three times subcultivation on the WPM medium with agar-agar content in concentration of 0.8%, 1%, and 1.2%. Experiment C included treatment of the isolated explants with osmotic liquid (OF) (Murashige and Skoog medium (MS) [52] with 2 M glycerol, 0.3 M sucrose and 0.2 M maltose) followed by 30 min incubation at room temperature. This experiment used explants of axillary buds obtained from microshoots cultivated on the WPM growth medium with agar-agar 0.8%. In all versions of the experiments, 3 weeks old plant culture was used for cryopreservation. Three days before manipulations, plants of all types of treatments were transferred to conditions of +4 °C and photoperiod of 16 h. Axillary buds were used as explants in cryopreservation in all versions, and they were extracted together with the stem and petiole tissues. Explants were 2–3 mm in size. PVS2 solution (30% glycerol, 15% ethylene glycol and 15% DMSO in liquid WPM medium containing 0.4 M sucrose; [53] was used for vitrification in all types of experiments; explants were transferred into this solution and incubated for 10 min at room temperature. PVS2 solution was removed with a Pasteur pipette and replaced with a new one, and after 20 min vials with explants and solution were cryopreserved by ultrafast cooling with liquid nitrogen LN (−196 °C). Test vials with samples of experiments were stored for 3 months. Subsequently, best options of the pretreatment conditions, which showed the higher survival rate, were repeated for all wild and transformed lines. In this case, samples were stored for 3 and 6 months.

### 4.3. Thawing and Recovering

Cryotubes of all types in the experiments were transferred to a water bath with 45 °C for 1 min. After thawing, PVS2 was drained from cryotubes and replaced with liquid MS medium containing 1.2 M sucrose [53], where the shoot tips were washed for 20 min at room temperature. Then, explants were placed on the 1/2 MS medium with 3% sucrose, 1 mg/l gibberellic acid and 0.8% agar-agar. After 2 days samples are transferred to recovery WPM medium containing 0.5 mg /l zeatin, 1 mg/l gibberellic acid and 0.8% agar-agar. Later incubation was taking place at 22–24 °C temperature and photoperiod of 16 h until a formed shoot appeared. Percentage of axillary bud survival rate and healthy shoot formation from the surviving meristems was collected after 3 weeks. Presence of morphological deviations in microshoots obtained from regenerants was assessed after 8 weeks.

### 4.4. Analysis of Genotype Preservation after Cryostorage

The approaches including variability molecular analysis and biometric analysis were used to determine the genotype integrity. The molecular analysis included determination of the recombinant construct presence in the transformed lines, as well as molecular microsatellite marking (SSR) of all lines after storing for 6 months in LN. Plants of all genotypes not exposed to cryostorage were used as control plants to analyze variability. DNA was isolated from all lines using DNeasy^®^Plant Mini kit (QIAGEN, Hilden, Germany) for molecular analysis. PCR was performed using the Encyclo Plus PCR kit (Evrogen, Moscow, Russia). Previously selected primers and conditions were used to determine the presence of *bar* and *sp-Xeg* construct [35,50]. Primers and conditions described for the *PTR2, PTR3, PTR4, PTR5, PTR6, PTR14* loci and selected for genotyping *P. tremuloides* were used in microsatellite marking [48,49]. For genotyping each locus, selected pair of primers was used consisting of forward primer marked with 6-carboxyfluorescein (6-FAM) and of unmarked downstream primer (Syntol Comp., Moscow, Russia). The amplification was performed on the MJ Mini thermal cycler (Bio-Rad Laboratories, Inc., Hercules, CA, USA). The fragment analysis was carried out by capillary electrophoresis on the ABI 3130xl Genetic Analyzer sequencer (Applied Biosystems, Foster, CA, USA). The S450 LIZ size standard (Syntol Comp. Moscow, Russia) was used as marker fragments. The peak identification and fragment sizes were determined using the Gene Mapper v4.0 software (Applied Biosystems, Foster, CA, USA).

After assessing the survival rate after cryopreservation, part of the microshoots 4–5 weeks old 1.5–2 cm in size of the Pt, Xeg-1-1a, Xeg-1-1c, Xeg-2-1b and Xeg-2-3b genotypes were excised and rooted on the WPM medium at 22–24 °C and at photoperiod of 16 h to analyze the interclonal variation. The same genotypes, but not cryopreserved, were used as control species. Rooted plants were transplanted into protected ground conditions (Biotron station, BIBCh, Pushchino, Russia). Height of the plants was measured after 6 months. Plants of Pt genotypes not exposed to cryostorage were used as control plants to analyze genetic variability under greenhouse conditions.

### 4.5. Data Collection and Statistical Analysis

Each experiment with each cryopreservation line was repeated twice and consisted of at least three repetitions per treatment, 18–20 samples each. Plants were transferred to greenhouse conditions with an amount of 20 plants per genotype. Statistical analysis of percentages was carried out by the test for homogeneity of proportions, and significant treatment differences selected by a non-parametric statistical test, the Post Hoc Multiple Comparisons Test [54].

## 5. Conclusions

Thus, we developed a technology that makes it possible to successfully preserve the wild aspen genotypes and the transformed lines with no changes in genetic status of culture. Average recovery rate in this technology is high reaching 83%. Such high percentage of survival rate is achieved due to the following technology features: preliminary cultivation on a denser growth medium; selection of axillary bud’s explants, obtained after removal of the apical bud; explants pretreatment with osmotic fluid; using the one-step freezing method and PVS2; the fast thawing rate. Usage of complex pretreatment made it possible to increase the regenerant survival rate from 55% to 83%. Genetic and morphological analysis showed that cryopreservation technology developed by us is the most suitable in long-term cryopreservation of aspen plants and does not lead to variation in the plant genotype. Our data demonstrated that survival rate of the explants could directly depend on specific genetic transformation that was carried out for a given genotype. It is important to take this into consideration in the process of cryopreservation. The results obtained allow us to conclude that the technology developed by us can be acceptable for long-term storage of valuable aspen genotypes. However, further genetic, histological and morphological studies are needed for a deeper understanding. We also plan to test this technology on other types of woody deciduous plants, in particular valuable birch genotypes.In this regard, in order to standardize and possibly increase the survival rate of regenerates, we plan to develop a cryostorage protocol using artificial intelligence models and optimization algorithms in plant cell and tissue culture. This approach has shown itself to be promising for various plants [55,56]. Thus, future studies will be able to use this strategy to predict and optimize cryopreservation methods for in vitro cultures of valuable woody plant genotypes.

## Figures and Tables

**Figure 1 plants-10-00077-f001:**
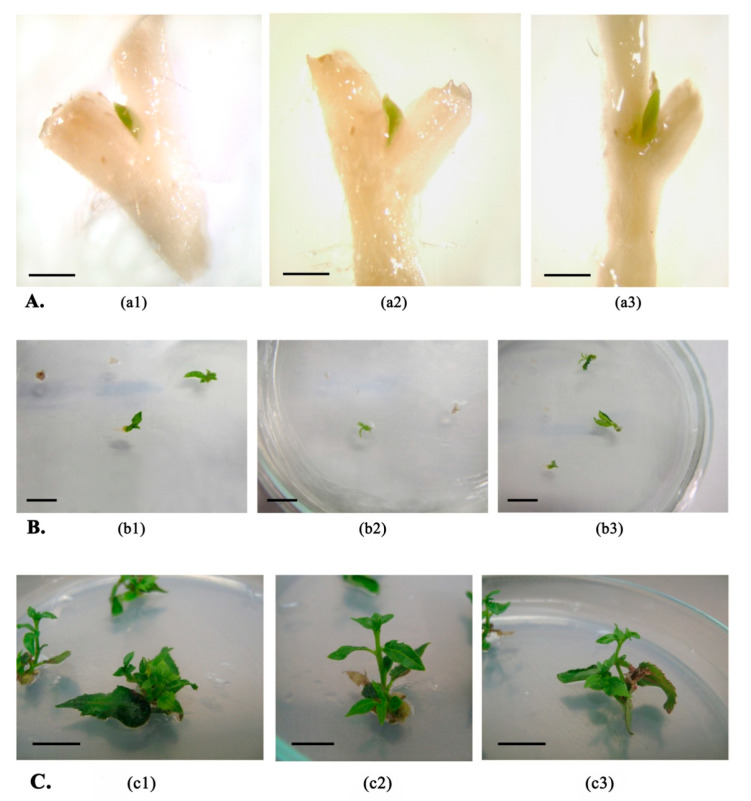
Aspen plant regenerants at different stages of recovery after cryopreservation. (**A**). Micrographs of the Pt genotype regenerants one week after transfer to recovery medium (bar: 0.5 mm); (**a1**) after changing the explant type (experimental conditions A); (**a2**) after preliminary cultivation on medium with agar-agar content of 1% (experimental conditions B); (**a3**) after using osmotic pretreatment (experimental conditions C). (**B**). Regenerants of the Pt genotype 3 weeks after transfer to recovery medium (bar: 1 cm); (**b1**) after changing the explant type (experimental conditions A); (**b2**) after preliminary cultivation on medium with agar-agar content 1.2% (experimental conditions B); (**b3**) after using osmotic pretreatment (experimental conditions C). (**C**). Regenerants of the Pt genotype 8 weeks after transfer to recovery medium (bar: 1 cm); (**c1**) after changing the explant type (experimental conditions A); (**c2**) after preliminary cultivation with agar-agar 1% content (experimental conditions B); (**c3**) after using osmotic treatment (experimental conditions C).

**Figure 2 plants-10-00077-f002:**
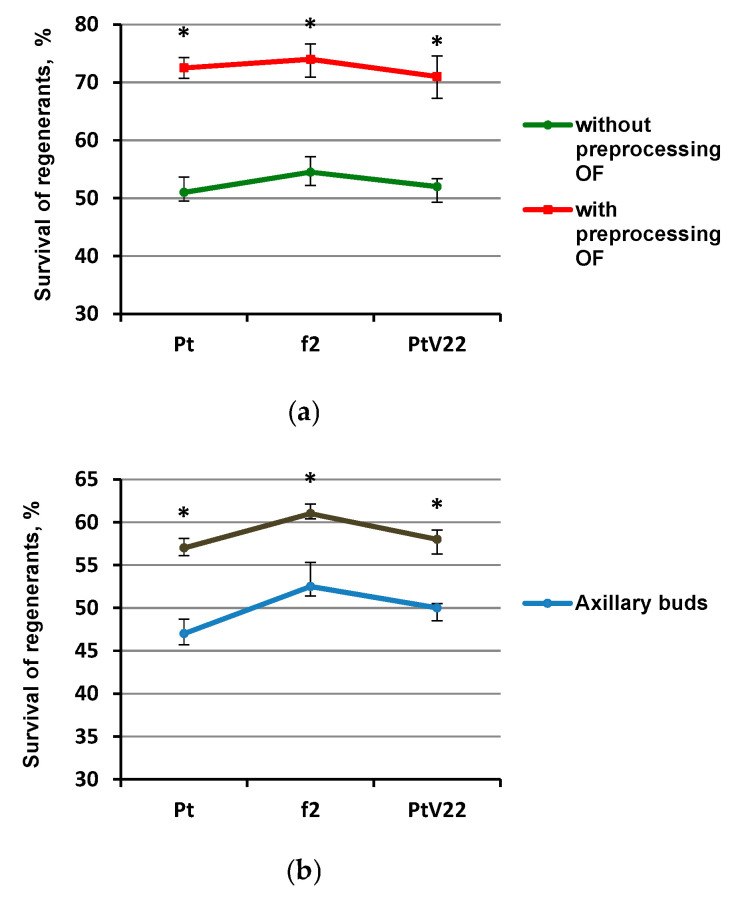
Regenerant survival dependence after incubation in LN for 3 months after different pretreatments. (**a**) Experimental conditions C (using osmotic treatment); (b) Experimental conditions A (change of explant type); (**c**) Experimental conditions B (pretreatment on medium with different density); * - percentages differs significantly at *p* ≤ 0.05 from percent with and without OF pretreatment.

**Figure 3 plants-10-00077-f003:**
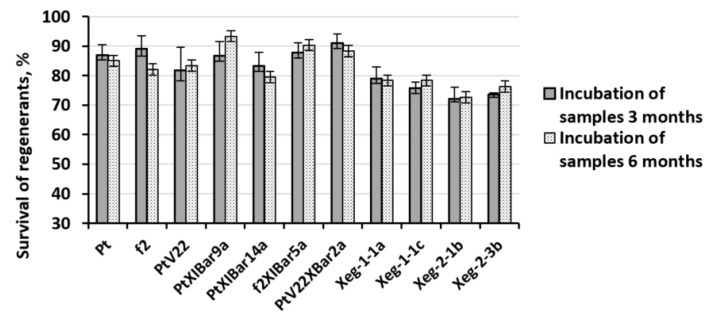
Survival rates of regenerants of transformed and non-transformed aspen lines after 3 and 6 months of incubation in LN. Regenerants were exposed to preliminary cultivation on growth medium with 1% agar-agar, removal of the apical bud and treatment with the OF.

**Figure 4 plants-10-00077-f004:**
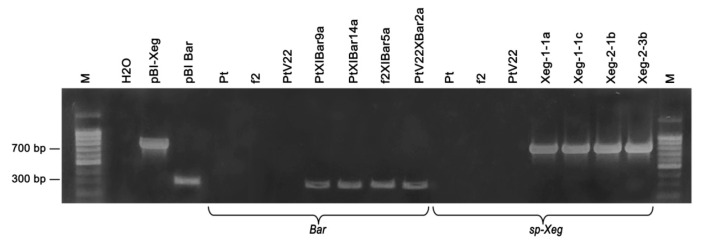
PCR analysis of the transgenic aspen lines after 6 months of cryostorage in LN for the presence of *bar* and *sp-Xeg* recombinant genes; *Bar* gene (expected amplicon size ~310 bp), *sp-Xeg* (expected amplicon size ~720 bp); M—DNA length marker 100 bp (Evrogen); H_2_O—negative reaction control, pBI-Xeg and pBIBar—plasmid DNA (positive control); Pt, f2 and PtV22—non-transgenic lines; transgenic lines: PtXIBar9a, PtXIBar14a, f2XIBar5a, PtV22XBar2a, Xeg-1-1a, Xeg-1-1c, Xeg-2-1b, Xeg-2-3b.

**Figure 5 plants-10-00077-f005:**
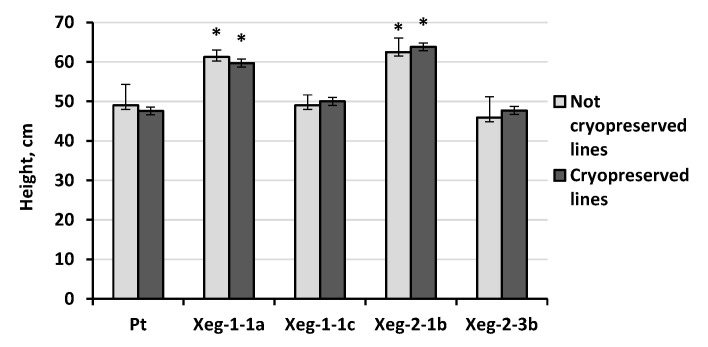
Average height of experimental plants exposed to freezing-thawing and incubation in LN, and control plants after 6 months of growing in greenhouse conditions. * - significantly different from Pt at *p* ≤ 0.05.

**Table 1 plants-10-00077-t001:** Allelic constitution of aspen plants by eight SSR loci.

Lines	Sample Status	PTR2	PTR3	PTR4	PTR5	PTR6	PTR14
Pt	C	204/204	197/201	215/212	258/258	247/243	163/159
	F	204/204	197/201	215/212	258/258	247/243	163/159
f2	C	204/204	195/195	217/196	258/258	247/243	181/162
	F	204/204	195/195	217/196	258/258	247/243	181/162
PtV22	C	204/204	197/195	215/197	257/257	243/243	176/156
	F	204/204	197/195	215/197	257/257	243/243	176/156
PtXIBar9a	C	204/204	197/201	215/212	258/258	248/243	165/156
	F	204/204	197/201	215/212	258/258	248/243	165/156
PtXIBar14a	C	204/204	195/195	215/212	258/258	247/243	163/159
	F	204/204	195/195	215/212	258/258	247/243	163/159
f2XIBar5a	C	204/204	195/195	196/196	258/258	247/243	181/162
	F	204/204	195/195	196/196	258/258	247/243	181/162
PtV22XBar2a	C	204/204	197/195	215/197	257/257	243/243	176/156
	F	204/204	197/195	215/197	257/257	243/243	176/156
Xeg-1-1a	C	204/204	197/201	215/212	258/258	247/243	177/161
	F	204/204	197/201	215/212	258/258	247/243	177/161
Xeg-1-1c	C	204/204	197/197	215/212	258/258	243/226	163/159
	F	204/204	197/197	215/212	258/258	243/226	163/159
Xeg-2-1b	C	204/197	199/197	215/212	257/258	247/243	163/156
	F	204/197	199/197	215/212	257/258	247/243	163/156
Xeg-2-3b	C	204/204	197/197	215/212	258/258	248/243	163/159
	F	204/204	197/197	215/212	258/258	248/243	163/159

C—control line, not cryopreserved; F—line cryopreserved for 6 months in LN.

## Data Availability

The data presented in this study are available on request from the corresponding author.

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
