# Peer review of "Efficient Cryopreservation of Populus tremula by In Vitro-Grown Axillary Buds and Genetic Stability of Recovered Plants"

_plants, 2021, doi:10.3390/plants10010077_

Round 1
Reviewer 1 Report
The ms describes the development of cryopreservation protocol for in vitro material of aspen, with good regeneration results. Furthermore, genetic fidelity of the cryopreserved material was studied and no variations were found. The experiments are well planned and sound, and results presented clearly. The genetic fidelity of cryostored materials is, however, not so understudied area as the authors let the reader to understand. There have been several publications on genetic fidelity issues during cryopreservation, including Populus too. So the novelty of the present study is not as high as one could assume based on the present introduction. I would anyhow recommend acceptance of the ms for publication, if only the following points are taken into consideration and revised accordingly:
Abstract
- r15 wild type lines
- r19 please use "pretreatment" to describe pretreatments throughout the ms instead of "preliminary use / preparation"
Introduction
- r53 PVS2 -based vitrification might the most acceptable method for cryopreservation of IN VITRO CULTURES of woody plants, but slow cooling of naturally cold acclimized buds works very well and is also very cost effective method - so please add " in vitro cultures"
- r85 It is not true that there is a complete lack of works regarding genetic fidelity of cryopreserved plant material ! This needs to be revised and references cited correctly. See e.g. Barra-Jimenez et al. 2015, Krajnakova et al. 2011, Aronen et al., Jokipiii et al. 2004
Results
- figures: please organize explanations for your different-coloured lines in Fig.3. in the same order as in which they appear in the figure, would make it easier for the reader; Figures 1, 2 and 4 could be combined into one photo plate
- Fig 6: PCR results do not tell the copy number of the transgenes, only their presence. Did you check the copy number ?
Discussion
- r208-209 You write: "At the same time, previously developed cryopreservation technologies for the Populus genus plants were not focused on saving the original genotype." How can you claim this ? Very difficult to imagine that anyone would have developed a cryo protocol in order to have anything else but the original genotype! Remembering also that some authors have also tested the genetic fidelity.. Please revise and respect earlier works too.
- Please add discussion on your genetic fidelity results in light of previously published works
- r286-87 you write: "This proves that allelic constitution of aspen plants which don’t survive cryopreservation is different." How do you know this if you did not analyze the explants that did not survive ??? And more over, when having clonal material, it is expected that all the plants have the same genotype, including both survived and dead explants.. Or are you claiming that some specific genotypes carrying certain alleles don't survive cryopreservation ?? How can you claim this without any evidence ? Revise this part.
M & M
- why did you use MS medium in pretreatment and thawing when the material was otherwise cultivated with WPM ? Please explain.
- r326 verification - do you mean vitrification ?
- r362 Did you really graft the microshoots ? This is very laborious indeed. WHY did you do that ? Why not to root them directly ?? Or is this a spelling error and you mean that they were excised and rooted ?
Reviewer 2 Report
There is virtually no novelty in this paper, whether from a scientific or technological perspective. The English should be considerably improved. I recommend rejection of the MS.Author Response
Please see the attachment

Reviewer 3 Report
The topic is very attractive, and I believe that the manuscript needs minor revision to completely meet the standards of the Journal.
Line 36: it needs reference(s)
Line 39: it needs reference(s)
Line 48: it needs reference(s)
Line 62: it needs reference(s)
The conclusion section is very short. At least it should discuss more future work. I suggest the following discussion with its citations (However, I only selected the newest papers and authors can provide more references.):
Machine learning algorithms provide a complementary prospect for fine-tuning protocol development, as they allow to predict optimal requirements in terms of incubation conditions, plant growth regulators, explant source, and genotype, without the need for large-scale, time-consuming, and costly experimental trials (Application of artificial intelligence models and optimization algorithms in plant cell and tissue culture. Appl Microbiol Biotechnol (2020), 104:9449–9485). The potential of this approach is illustrated by recent reports on the application of different machine learning algorithms to predict and optimize secondary metabolite production (Optimization of salicylic acid and chitosan treatment for bitter secoiridoid and xanthone glycosides production in shoot cultures of Swertia paniculata using response surface methodology and artificial neural network. BMC Plant Biol. 2020, 20, 225.; Combining medicinal plant in vitro culture with machine learning technologies for maximizing the production of phenolic compounds. Antioxidants 2020, 9, 210.), somatic embryogenesis (Introducing a hybrid artificial intelligence method for high-throughput modeling and optimizing plant tissue culture processes: the establishment of a new embryogenesis medium for chrysanthemum, as a case study. Applied Microbiology and Biotechnology, (2020) 104(23), pp.10249-10263.), shoot proliferation (Optimization of culture conditions for differentiation of melon based on artificial neural network and genetic algorithm. Sci Rep 10(1):3524.; Use of multiple regression analysis and artificial neural networks to model the effect of nitrogen in the organogenesis of Pinus taeda L. Plant Cell Tiss Org 137(3):455–464.), and hairy root culture (A neural network approach for the prediction of in vitro culture parameters for maximum biomass yields in hairy root cultures. J Theor Biol 265(4):579–585.; Envisaging the regulation of alkaloid biosynthesis and associated growth kinetics in hairy roots of Vinca minor through the function of artificial neural network.Appl BiochemBiotechnol 178(6):1154–1166.). Therefore, future studies can use this strategy for predicting and optimizing in vitro tissue culture and cryopreservation of this valuable plant.
